# Effects of Nutritional Interventions on Accuracy and Reaction Time with Relevance to Mental Fatigue in Sporting, Military, and Aerospace Populations: A Systematic Review and Meta-Analysis

**DOI:** 10.3390/ijerph19010307

**Published:** 2021-12-28

**Authors:** Liam S. Oliver, John P. Sullivan, Suzanna Russell, Jonathan M. Peake, Mitchell Nicholson, Craig McNulty, Vincent G. Kelly

**Affiliations:** 1School of Exercise and Nutrition Sciences, Queensland University of Technology, Brisbane, QLD 4059, Australia; liamscott.oliver@hdr.qut.edu.au (L.S.O.); jonathan.peake@qut.edu.au (J.M.P.); mitchell.nicholson@hdr.qut.edu.au (M.N.); c.mcnulty@qut.edu.au (C.M.); 2The Brain Always Wins, LLC, Newport, RI 02840, USA; jps@performancedocs.com; 3School of Behavioural and Health Sciences, Australian Catholic University, Brisbane, QLD 4014, Australia; Suzanna.Russell@acu.edu.au; 4School of Biomedical Sciences, Queensland University of Technology, Brisbane, QLD 4059, Australia; 5School of Human Movement and Nutrition Sciences, University of Queensland, Brisbane, QLD 4059, Australia

**Keywords:** mental fatigue, sport, military, aerospace, nutrition, intervention, ergogenic

## Abstract

*Background*: Research in sport, military, and aerospace populations has shown that mental fatigue may impair cognitive performance. The effect of nutritional interventions that may mitigate such negative effects has been investigated. This systematic review and meta-analysis aimed to quantify the effects of nutritional interventions on cognitive domains often measured in mental fatigue research. *Methods*: A systematic search for articles was conducted using key terms relevant to mental fatigue in sport, military, and aerospace populations. Two reviewers screened 11,495 abstracts and 125 full texts. A meta-analysis was conducted whereby effect sizes were calculated using subgroups for nutritional intervention and cognitive domains. *Results*: Fourteen studies were included in the meta-analysis. The consumption of energy drinks was found to have a small positive effect on reaction time, whilst the use of beta-alanine, carbohydrate, and caffeine had no effect. Carbohydrate and caffeine use had no effect on accuracy. *Conclusions*: The results of this meta-analysis suggest that consuming energy drinks may improve reaction time. The lack of effect observed for other nutritional interventions is likely due to differences in the type, timing, dosage, and form of administration. More rigorous randomized controlled trials related to the effect of nutrition interventions before, during, and after induced mental fatigue are required.

## 1. Introduction

Mental fatigue is referred to as a psychobiological state resulting from prolonged periods of demanding cognitive activity [1]. Research on the effects of mental fatigue has increased exponentially in sport and exercise science literature in the last decade [2]. The effects of mental fatigue have also been investigated in relation to military [3] and aerospace [4] environments. Mental fatigue has been described by professional footballers as a difficulty in maintaining concentration, impaired reaction times, feelings of sleepiness, and reduced motivation [5]. Elite male cyclists reported that a 30 min (min) mentally fatiguing computerized task required considerable mental effort and increased feelings of frustration [6]. Similarly, soldiers embarking on a 51 h sustained operation reported greater feelings of mental fatigue [7], whilst 94.1% of airline pilots experienced subjective mental fatigue [8]. Mental fatigue has typically been measured using subjective scales such as a 100 mm (mm) visual analog scale (VAS), questionnaires (e.g., Profile of Mood States (POMS), the Brunel Mood Scale (BRUMS)), and cognitive performance tasks [9]. These methods have likely been used because of their practicality; however, they require validation regarding the assessment of mental fatigue. Furthermore, there are significant conceptual and operational challenges to mental fatigue and ego depletion [10,11,12,13,14] that require elaboration. Several models of fatigue have been proposed, such as the central governor theory [15], the psychobiological model [16], and a complex systems approach to exercise-induced fatigue [17]. These models describe fatigue as an interaction of central and afferent feedback and subsequent action, and systems methods treat the system as a unit of analysis, rather than focusing on human behavior alone [18].

Behaviorally, mental fatigue may impair technical, tactical, and decision-making abilities in sport [19]. Induced mental fatigue has been shown to impair cognitive performance (e.g., response time and accuracy) in some sport- and exercise-based studies [20] but not others [5,21,22]. This may be because computerized cognitive tasks are often used to assess cognitive performance, and such tasks likely do not measure the multiple cognitive domains that interact during physical, sport-specific situations. Indeed, it has been suggested that mental fatigue could impair cognitive domains such as response time and accuracy in sport-specific psychomotor tasks including soccer-specific decision making [23] and badminton-based visuomotor tasks [24]. Research in military populations has also shown that mental fatigue resulted in impaired marksmanship decision accuracy [25]. Furthermore, fatigue and sleepiness increase the risk of accident occurrence and ultimately harm the safety of aircraft pilots [26]. Electroencephalography measures have highlighted that the induction of mental fatigue impairs neurotransmission and increases alpha and theta band activity in the anterior cingulate cortex and prefrontal cortex of the brain [27]. These brain regions are associated with working memory [28], decision making [29], and working memory and emotion [30]. It is plausible to suggest that impairments in these cognitive domains may be indicators of mental fatigue.

Cognitive domains have primarily been defined and categorized with respect to the assessment of clinical conditions associated with cognitive impairment, such as the fifth edition of the Diagnostic and Statistical Manual of Mental Disorders [31] and Cattrell–Horn–Carroll [32] models of cognition. However, these models do not fully address the complexity and relevance of cognitive domains in general, as well as within sport, military, and aerospace populations. These domains require high levels of cognitive effort to sustain attention and react rapidly to unpredictable stimuli, be they an opponent in team sports, sustained operations in the armed forces, or changing weather conditions for aircraft pilots. One model that discusses cognitive domains that are transferable to sport, military, and aerospace contexts is the model developed by Harvey [33]. The Harvey [33] model outlines eight broader cognitive domains such as motor skills and attention and concentration, as well as their subdomains. In addition, Harvey [33] refers to functions such as motor skill learning, visual function, and automaticity, which arguably all hold relevance to sport, military, and aerospace contexts. Mental fatigue research has often measured cognitive performance with reference to the cognitive domains assessed by a given task, such as executive function in the Stroop task or reaction time in the psychomotor vigilance task (PVT). However, these tasks have not been discussed with a defined model of cognition as context. It suffices to say that mental fatigue may be a product of a combination of factors such as the regulation of affect, the availability of energy, neurotransmission, and the timing and dosing of load [34], all of which are highly relevant to mental fatigue research. The management of response inhibition is also a key component of executive functioning, and response inhibition is often discussed with regard to accuracy and reaction time in mental fatigue research in the Stroop task [9] due to the need to differentiate between correct and incorrect stimuli quickly. Simpler tasks challenge reaction time as the dominant domain, such as the PVT which requires a rapid reaction to a stimulus by pressing a key in response to the appearance of a flashing marker in computerized tasks. Accuracy and reaction time are, therefore, performance outcomes often measured in mental fatigue research [20,22] and could be discussed with reference to Harvey’s domains [33].

Nutritional interventions have been investigated to determine their influence on individual responses to cognitive demand in sport and exercise science [35], military [36], and aerospace [37] studies. The administration of nutritional interventions after a mentally fatiguing intervention, e.g., psychostimulants such as caffeine [38] and carbohydrate [22], has been shown to enhance physical performance and reduce ratings of perceived exertion (RPE) during subsequent exercise and improve mood by increasing feelings of vigor [38]. Accepting that mental fatigue has the potential to reduce aspects of performance, increase RPE, and impair mood, the same nutritional interventions could theoretically mitigate the negative effects of mental fatigue. Although not specific to mental fatigue, beta-alanine ingestion has been shown to improve executive function, measured as the time to complete a Stroop task [39]. Within research specific to mental fatigue, consuming 5 mg·kg^−^^1^·BM^−1^ caffeine was shown to improve 20-km time trial performance in 12 nonprofessional trained cyclists [40]. Similarly, Van Cutsem et al. [22] showed that a caffeine–maltodextrin mouth rinse reduced subjective ratings of mental fatigue, yet cognitive performance was unaffected. In the military environment, caffeinated chewing gum has been shown to positively affect soldiers’ shooting performance and vigilance [41,42]. Furthermore, soldiers’ vigilance improved when in a sleep-restricted state [43]. However, in rugby players, repeated sprint performance and cognitive ability were unaffected after chewing caffeine gum [44]. Similarly, during simulated flight procedures, pilots’ cognitive and psychomotor performance was improved in those who were not well rested, compared with those who were [4], nuancing the interpretation of current research. In addition, a recent position on beta-alanine posited that initial results in tactical athletes showed a positive effect on military-specific tasks [45]. Evidently, there are several options for nutritional interventions, with forms of administration, dosages, timings, types, and mechanisms of action. Nonetheless, nutritional interventions have been recognized as a practical and potentially effective method to enhance cognitive performance.

Research has investigated the influence of several nutritional interventions such as carbohydrate and caffeine supplementation on mental fatigue using objective and subjective measures in sport, exercise, military, and aerospace settings. However, the most effective nutritional intervention to negate any potential negative of mental fatigue remains unknown. Therefore, systematically reviewing and meta-analyzing the effects of nutritional interventions on cognitive outcomes typically affected by mental fatigue could support the formation of guidelines surrounding nutritional intake to reduce the risk of suffering cognitive impairment and, thus, inform practitioners’ decision making in sport, military, and aerospace contexts. This systematic review and meta-analysis aims to quantify the effects of nutritional interventions on cognitive performance outcomes that are commonly measured in mental fatigue research and subsequently identify the most effective nutritional intervention to mitigate the potential impairments in the context of sport, exercise, military, and aerospace research.

## 2. Materials and Methods

### 2.1. Information Sources and Search Strategy

PubMed, Scopus, SPORTDiscus, and Google Scholar were used to search for articles published from 2000–2020 based on the breadth of mental fatigue research in sport and exercise science [9,20,46]. This timeframe was chosen to maximize the likelihood of finding studies (1) with similar methods of assessing cognitive domains, (2) which could be discussed according to a model of cognition supported by recent neuropsychological evidence [33], and (3) which could apply to sport, military, and aerospace contexts. All results were screened except for those found from Google Scholar, from which a maximum of the first 1000 articles were screened. Search terms are shown in Table 1.

### 2.2. Eligibility Criteria

The eligibility criteria followed an agreed PICOS (Participants, Intervention, Comparators, Outcomes, Study design) that was uploaded to https://osf.io/6rshw/ (code uploaded created on 31 October 2021).

### 2.3. Participants

Studies were included if participants were healthy (no history of clinically diagnosed health conditions such as cognitive impairments or disorders, e.g., dementia) with a mean age of 18–45 years old. Exclusion criteria covered animals, humans with clinically diagnosed mental or physical conditions or unclear health status, and samples whereby all the criteria were not met or clearly described.

### 2.4. Interventions

Studies that employed a nutritional intervention whilst measuring any element of mental fatigue were included. Nutritional interventions included any dietary supplementation strategy or manipulation administered alongside the assessment of mental fatigue or any of its derivatives (e.g., self-control, cognitive fatigue) as an outcome (e.g., brain activity or subjective mental fatigue). This included aspects such as working memory and response inhibition which are likely impaired by induced mental fatigue [9]. Studies were required to have clearly stated some relation to cognitive performance or sport and exercise performance as per the cognitive domains discussed in the Harvey model [33]. Moreover, studies must have clearly shown implications for sport, military and aerospace populations and must have clearly articulated some relation to mental fatigue and any of its derivatives. The inclusion of a mentally fatiguing intervention of any type and duration, such as the Stroop task [22] or AX-continuous performance task [9], was optional and was recorded. Studies were also accepted if they measured mental fatigue in some way, such as using a VAS or brain activity. A manipulation check, such as significantly impaired performance from the beginning to the end of the mentally fatiguing task, was also accepted as an indicator of the induction of mental fatigue.

### 2.5. Comparators

Comparators included any placebo or no-supplement condition that showed significantly lower subjective mental fatigue (e.g., visual analog scales or questionnaires), alongside any other objective markers (statistical significance at *p* < 0.05).

#### Main Outcome(s)

The effects of any nutritional intervention on subjective and objective markers related to mental fatigue, sport and exercise performance, changes in brain activity, or any other measures taken were included, provided that the authors clearly stated the relation to mental fatigue or its derivatives. Any study that failed to meet all inclusion criteria was excluded. Once studies were identified and selected, the reference lists were screened to ensure no eligible studies were omitted. Other reviews and meta-analyses relevant to the topic were also screened to identify any potentially overlooked studies. Where full texts were unavailable, corresponding authors were contacted to determine whether the study was published, and the full text was ultimately requested. Conference abstracts and unpublished studies were excluded.

### 2.6. Study Design

Randomized controlled trials and nonrandomized controlled trials written in English were included. Crossover and parallel-group design studies were eligible.

### 2.7. Study Selection and Data Extraction

Two reviewers (L.S.O. and M.N.) independently screened the titles and abstracts of articles during the first stage of study selection using the Abstrackr [47] website, guided by a PICO(S) statement of criteria agreed by all authors. Abstrackr was used according to the guidelines of Polanin et al. [48]. During the second stage, full texts of each included article were screened independently by the primary reviewers. Conflicts at both stages were resolved by discussion and consensus between L.S.O., M.N., and V.G.K. Where data were unavailable after being included in either stage, attempts were made to contact the authors of these studies to retrieve the data.

### 2.8. Study Quality Assessment

The Downs and Black [49] scale was adapted and used to determine the quality of the individual studies. This comprised five subscales containing items related to reporting, validity, bias, confounding, and power. Answers were scored in alignment with the recommendations of Downs and Black [49]; items 9, 13, 16, and 17 were removed, and the wording of item 5 was changed to link more closely to the aims of this study (Appendix A). Independent assessment was on a study level by L.S.O. and M.N. Any conflicting viewpoints were resolved by SR.

### 2.9. Meta-Analysis

Initially, effect sizes were calculated at an individual study level as most studies reported multiple outcomes; however, some only reported one outcome. For studies with multiple outcomes, the effect size of each outcome was calculated and pooled to create one overall study effect size. All analyses were conducted in R version 4.0.5 (RStudio environment version 1.2.5001, Boston, MA, USA) using the dmetar [50], meta [51], and metafor [52] packages. The inverse variance method was used, and the α for all tests was set at 5%. Between-study heterogeneity was assessed using the restricted maximum likelihood estimator for τ^2^ [52], and the Q-profile method was used to obtain the confidence interval of τ^2^ and τ. Hartung–Knapp adjustments were applied if appropriate. Once the meta-analysis outcomes within one study were calculated, these were compiled into their respective subgroups. The codes that support the findings in this article were uploaded at https://osf.io/6rshw/ (code uploaded on 31 October 2021). Meta-analyses were conducted using guidance from an open-source repository of Harrer et al. [50].

### 2.10. Data Synthesis

The pooled data of the outcome measures from each study were reported using a random effects model to control for inter-study heterogeneity, and to report on standardized mean differences (SMDs). The immediate pre-nutritional intervention assessment was compared with the first post-intervention assessment to standardize the analysis. Given the differences in outcome measurements, imputed standard deviations (*S_imput_*) were calculated using mean change between experimental and control conditions for crossover design studies, whilst the difference between the mean change of experimental and control groups was calculated for parallel design studies. A correlation coefficient of 0.5 was used to calculate *S_impu_*_t_ and confidence intervals in the meta-analysis, which has been reported to be acceptable when including studies with missing data [53]. Thus, SMDs were calculated with the following equation:(1)$$S_{imput\;}=\sqrt{\left(\left(baselineSD^{2}\right)+\left(postSD^{2}\right)+\left(2\times 0.5\times baselineSD\times postSD\right)\right)\;}$$

### 2.11. Analysis of Subgroups

Effect sizes for nutritional interventions were subgrouped once the results of the systematic review determined which could be included in the meta-analysis. Subgroups for cognitive domains were assigned by J.P.S. and L.S.O. using the Harvey [33] domains before data analysis. Within the meta-analysis, these were further divided into reaction time and accuracy, as deemed appropriate from the available data. To capture a larger dataset, different doses were compared with placebo (e.g., 3 mg⸱kg^−1^⸱BM^−1^ and 6 mg⸱kg^−1^⸱BM^−1^ caffeine versus placebo).

## 3. Results

The searches yielded 14,955 studies and, after deduplication, 11,319 abstracts were screened. A total of 125 full texts were screened, and 14 studies were included in the meta-analysis (Figure 1).

### 3.1. Included Studies

Study details and outcomes of the 14 studies are presented in Table 2. The nutritional interventions investigated in the included studies were beta-alanine (*n* = 2), caffeine (*n* = 6), carbohydrate (*n* = 4), and energy drinks (*n* = 2).

### 3.2. Study Quality

The average quality of evidence of the studies in this review, using the Downs and Black checklist [49], was 93% ± 5% (Figure 2).

### 3.3. Meta-Analysis

Fourteen studies comprising 26 effect sizes the following nutritional interventions were included in the meta-analyses: beta-alanine (*n* = 2), caffeine (*n* = 14), carbohydrate ingestion (*n* = 6), and energy drinks (*n* = 4). None of the studies identified used mentally fatiguing interventions, but all mentioned and/or were deemed to include measures of cognitive performance (i.e., reaction time and accuracy) that were commonly measured in mental fatigue research. The only performance outcomes that were closely related between the available studies were accuracy and reaction time. Results are reported as the SMD plus 95% confidence intervals (CI). It was deduced that the cognitive domain executive function could be assigned; however, given the vast differences in cognitive tasks used in the studies included, the meta-analysis refers to the specific cognitive subdomains of accuracy and reaction time. Accuracy and reaction time were used as performance outcomes. Results per study are summarized using forest plots in Figure 3 and Figure 4.

### 3.4. Accuracy

#### 3.4.1. Caffeine

Seven effect sizes from six studies showed that caffeine had no effect on accuracy (−0.16 [−0.41, 0.08]) [44,54,55,56,57,58].

#### 3.4.2. Carbohydrate

Data from two studies showed that carbohydrate-based strategies had no effect on accuracy (−0.12 [−0.39, 0.15]) [59,60].

### 3.5. Reaction Time

#### 3.5.1. Beta-Alanine

Two studies showed that changes in reaction time favored placebo compared with a beta-alanine condition to a small extent (−0.30 [−1.27, 0.66]) [61,62].

#### 3.5.2. Caffeine

Six effect sizes from five studies showed that caffeine had no effect on reaction time (0.08 [−0.15, 0.31]) [44,54,55,56,57,58,63].

#### 3.5.3. Carbohydrate

Data from four studies suggests that the use of carbohydrate strategies had no effect on reaction time (−0.04 [−0.09, 0.17]) [59,60,63,64].

#### 3.5.4. Energy Drinks

A total of four effect sizes from two studies, one from Lassiter et al. [65] and three from the study of Pomportes et al. [42], showed that the consumption of energy drinks had the greatest effect in favor of the intervention (0.64 [0.13, 1.16]).

**Table 2 ijerph-19-00307-t002:** Details of individual studies included in the meta-analysis.

Subgroup	Study	Methodology	Sample Demographics	Nutritional Intervention	Placebo/Control Comparison	Relevant Outcome/s	Cognitive Domain	Effects Size/s (95% CI)
Beta-alanine	Wells et al. [61]	Parallel groups	PLA: 10 SUP: 9	12 g sustained release beta-alanine × 14 days	12 g rice powder × 14 days	ANAM: SRT Code substitution Mathematical processing Code substitution—delayed	RT	0.31 (−1.29; 0.67)
	(Varanoske et al. [62]	Parallel groups	PLA: 10 SUP: 9	12 g sustained release beta-alanine × 14 days	12 g rice powder × 14 days	Serial Sevens Test RT (visuomotor) Visual tracking (multiple object tracking)	Accuracy	0.13 (−4.93; 4.67)
Caffeine	Ali 2015 [54]	RCT	10	6 mg·kg⸱BM^−1^ anhydrous caffeine (Fluka Sigma-Aldrich, St. Louis, MO, USA)	Artificial sweetener (Equal)	CRT Stroop task	Accuracy RT	0.11 (−2.39; 2.61) 0.44 (−0.20; 1.08)
	Baur et al. [55]	Parallel groups	PLA: 12 SUP: 14	400 and 200 g regular coffee (101 ± 0.6 mg caffeine·200 g)	Decaffeinated coffee (2.4 ± 0.05 mg caffeine·200 g)	PVT Visuospatial n-back task Letter n-back task Visual search task	Accuracy RT	−0.06 (−0.08; −0.04) −0.40 (−0.81; 0.01)
	Beaven 2013 [58]	RCT	21	240 mg caffeine with/without blue light	Visually indistinguishable sugar PLA with/without blue light	PVT	Accuracy RT	0.03 (−0.78; 0.84) 0.00 (−0.20; 0.20)
	dePauw 2017 [63]	RCT	11	Caffeine nasal spray (15 mg/mL) + HPMC (2 g) + mannitol (2.57 g)	Distilled water, benzalkonium chloride (40 mg), HPMC (2 g) and mannitol (2.57 g)	Stroop task	RT	0.06 (−0.17; 0.29)
	Duncan 2018 [56]	RCT	12	5 mg·kg⸱BM^−1^ caffeine capsules (Myprotein, Manchester, UK)	5 mg·kg⸱BM^−1^ dextrose capsules (Myprotein, Manchester, UK)	Flanker task	Accuracy	−0.41 (−1.10; 0.28)
	Karayigit 2020 [57]	RCT	17	0.16 g·kg⸱BM^−1^ caffeinated coffee (6 mg·kg^−1^ caffeine) 0.08 mg⸱kg^−1^⸱BM^−1^ caffeinated coffee (3 mg⸱kg^−1^⸱BM^−1^ caffeine) (Nescafé Gold, Nestlé, Istanbul, Turkey)	Equal volume decaffeinated coffee	Flanker task	Accuracy RT Accuracy RT	3 mg⸱kg^−1^⸱BM^−1^: 0.23 (−0.86; 1.32) 6 mg⸱kg^−1^⸱BM^−1^: −0.39 (−0.59; −0.19) 3 mg⸱kg^−1^⸱BM^−1^: 0.41 (−3.56; 4.38) 6 mg⸱kg^−1^⸱BM^−1^: 0.4 (0.06; 0.84)
	Russell 2020 [44]	RCT	14	Caffeine gum (400 mg; 4.1 ± 0.5 mg⸱kg^−1^⸱BM^−1^)	Four pieces of PLA chewing gum	SRT Stroop task	Accuracy RT	0.37 (−0.21; 0.95) 0.16 (−0.58; 0.90)
Carbohydrate	de Pauw 2017 [63]	RCT	11	Glucose nasal spray (80 mg·mL^−1^) with HPMC (2 g) and mannitol (2.57 g)	Distilled water, benzalkonium chloride (40 mg), HPMC (2 g) and mannitol (2.57 g)	Stroop task	RT	0.03 (−0.50; −0.56)
	Harper 2017 [59]	RCT	15	60 g CHO + 205 mg NA^+^	PLA-electrolyte beverage or water (PLA beverage used as comparison in the meta-analysis)	COMPASS battery: Mean speed and accuracy scores (secondary and working memory, attention, and decision-making) Immediate and delayed word recall (# of correct responses)	Accuracy RT	−0.11 (−0.39; 0.17) 0.16 (−0.08; 0.40)
	Konishi 2017 [60]	RCT	8	25 mL mouth-rinse × 5 s 6.4% maltodextrin (Maltodextrin; Body Plus International, Miyagi, Japan; 6% CHO solution	Water	Stroop task (incongruent)	Accuracy RT	−0.19 (−1.17; 0.79) 0.43 (−0.56; 1.42)
	Lee 2014 [64]	RCT	12	68 g via a sports drink	6 kcal, 0 g CHO PLA drink	CRT PVT Symbol digit matching Search and memory Digit span	RT	−0.01 (−0.12; 0.10)
Energy drinks	Pomportes 2017 [42]	RCT	10	6% CHO (fructose [89%] and maltodextrin [11%]; Isoxan Sport Pro, NHS, Rungis, France) 200 mg CAF (CAF: Prolab Nutrition, Chatsworth, CA) added with orange sugarless syrup, 3.4 g GUAc (300 mg guarana + 100 mg ginseng + 180 mg vitamins C; Isoxan Actiflash Booster, NHS, Rungis, France)	Tap water added with orange sugarless syrup	Simon task	RT CHO: 0.53 (−0.36; 1.42) Caffeine: 0.62 (−0.28; 1.52) Guarana: 0.89 (−0.04; 1.82)	
	Lassiter et al. [65]	RCT	15	54 g CHO, 160 mg caffeine, 2 g taurine, 400 mg Panax ginseng, and 5 g of other ingredients	0 kcal beverage, trace ingredients (NR)	CRT Tapping task Go/No-Go (EF test) Stroop Test	RT −0.04 (−2.48; 2.40)	

## 4. Discussion

The aim of the study was to quantify the effect of nutritional interventions on cognitive parameters commonly measured in mental fatigue research in sport, military, and aerospace populations. The results indicated that energy drinks had a small significant positive effect on reaction time. None of the other nutritional interventions had a significant effect on accuracy or reaction time. Each individual supplement is discussed in further detail below.

### 4.1. Beta-Alanine

Two military studies were subgrouped to examine the effect of beta-alanine on reaction time [61,62] Wells et al. [61] randomly assigned 19 soldiers to groups of 12 g per day beta-alanine (*n* = 10) or placebo (*n* = 9) for 14 days before assessing performance in the Automated Neuropsychological Assessment Metrics (ANAM) during a 24 h sustained operations protocol. The ANAM is a clinically validated cognitive assessment tool, measuring functions such as reaction time and mathematical abilities [66], although beta-alanine had no effect on any of the ANAM assessments [66]. Using a subset of data from the study of Wells et al. [61], another study found that an identical supplementation protocol had no significant effect on difference in serial subtraction performance and visuomotor reaction time compared with placebo, although the placebo group had significantly more misses on a timed visuomotor task [62]. These findings could be a result of sustained operations, meaning that baseline measures (i.e., 0 h) were compared with measures taken 12 h post intervention. In the study of Smith et al. [9], despite reaction time in a 3 min PVT being impaired immediately after a mentally fatiguing AX-CPT and Stroop task, it recovered 30 min later, during which time participants read from emotionally neutral magazines. This suggests that a short period of time with little to no cognitive demand may be sufficient for reaction time to recover. Interestingly, in the study of Wells et al. [61], significant increases in serum monocyte chemoattractant protein-1 coincided with increased sleepiness at the 18 h and 24 h intervals, although Varanoske et al. [62] found no significant differences between groups for concentrations of biomarkers such as brain-derived neurotropic factor and C-reactive protein. These studies may have implications for the mechanisms of the effects of sleep restriction on cognition. It is difficult to differentiate between the effects of induced mental fatigue and sleepiness due to the limitations of controlling for variables such as the extent of sleep deprivation or induced mental fatigue. Although inferences can be discussed using only two studies, the results of this meta-analysis suggest that beta-alanine did not affect reaction time. More research is needed to reveal whether this could be the case immediately after the inducement of mental fatigue, both with and without sleep restriction. Future studies could assess cognitive performance alongside changes in brain activity at regions associated with induced mental fatigue, such as the prefrontal cortex and anterior cingulate cortex, as well as serum monocyte chemoattractant protein-1, to indicate whether mental fatigue and sleep restriction result in similar physiological changes.

### 4.2. Caffeine

The pooled effect size calculated in this meta-analysis suggests that caffeine had no effect on reaction time or accuracy. An experiment conducted by Beaven et al. [58] compared the effects of 240 mg of caffeine with 1 h of exposure to ~40 lux blue light. Twenty-one healthy individuals performed a computer-based PVT before and after one trial each comprising the following interventions: blue light plus caffeine, blue light plus placebo, white light plus caffeine placebo, and white light plus placebo. The blue light only and caffeine only conditions produced greater accuracy during a visual Go/NoGo and caffeine had an additive ergogenic effect on reaction time (i.e., faster responses with blue light plus caffeine). However, in terms of reaction time during the Flanker task, blue light consistently outperformed caffeine regarding the extent of performance enhancement. A plausible suggestion could be that common side-effects of caffeine intake such as greater anxiety could obstruct any performance-enhancing potential of caffeine’s stimulatory properties, although this could depend on the dosage of caffeine and genetic profile of the individual [67]. Therefore, non-nutritional interventions without said side-effects, such as blue light exposure, may be more appropriate for some individuals. In lieu of these findings, future research could create new knowledge by examining the effect of alternative interventions such as blue light exposure on cognitive performance in mentally fatigued individuals.

Relevant to the findings of Beaven et al. [58], Baur et al. [55] restricted homozygous C-allele carriers of adenosine A2A receptors to five nights of only 5 h time in bed. Participants then received either caffeinated (*n* = 12; 200 mg caffeine at breakfast and 100 mg caffeine after lunch) or decaffeinated coffee (*n* = 14) after sleep restriction, with cognitive performance (PVT, visual search task, and visuospatial and letter n-back tasks) measured at regular intervals. Response accuracy and speed decreased, while the number of lapses of attention increased in the PVT with increasing sleep restriction, and regular coffee significantly attenuated performance decline. C-allele carriers of ADORA2A receptors have been shown to benefit from improved cycling performance compared with T-allele carriers [68]. In addition, Baur et al. [55] found that there were no differences between any cognitive performance measures variables after a recovery night of 8 h in bed. It is possible that 8 h in bed resulted in sufficient sleep for participants to recover from potential impairments in cognitive performance. Further research is required to determine the effect of genetics on cognitive performance, and the time-course of recovery from induced mental fatigue, to provide novel data to current mental fatigue research.

In sport and exercise science research, Duncan et al. [56] examined the cognitive performance of 12 males after ingesting a 5 mg·kg^−1^·BM^−1^ caffeine or placebo capsule and found that response accuracy was significantly greater during incongruent trials, and response speed was significantly faster with caffeine during both congruent and incongruent tasks Elsewhere, 10 female athletes consumed a 6 mg·kg^−1^·BM^−1^ caffeine or placebo capsule and completed a Stroop task and choice reaction time task before, during, and after a 90 min intermittent treadmill-running protocol, to no significant effect [57]. A crossover design study tested the effects of 3 mg·kg^−1^·BM^−1^ and 6 mg·kg^−1^·BM^−1^ of coffee-derived caffeine on reaction time and response accuracy during a modified Flanker task, compared with decaffeinated coffee, in 17 female team sport athletes [57]. A congruent and incongruent version of the Flanker task was completed and compared with prior to ingestion; reaction time was significantly faster after 6 mg·kg^−1^·BM^−1^ post ingestion in the caffeinated coffee condition, with no difference between 3 mg·kg^−1^·BM^−1^ and placebo conditions. A noteworthy observation is that reaction time was significantly faster in the 6 mg·kg^−1^ condition after subsequent resistance exercise tests. Since these studies showed faster reaction times after exercise [56,57], it is possible that exercise has greater ergogenic potential than the use of a nutritional intervention on mental fatigue-related tasks. Research has suggested that completing a mentally fatiguing task for 30 min could reduce the amount of volume performed during a resistance training session [69]. Alternatively, muscular endurance protocols have been shown to improve executive function [70], and research could investigate the effectiveness of chronic resistance exercise interventions as a method of improving executive function and building tolerance to the negative effects of mental fatigue. Lastly, the cognitive performance of 14 professional academy rugby union players was assessed before and after chewing caffeinated gum [44]. Players chewed a placebo gum before completing repeated sprints, a Stroop task, and simple reaction time task. They then chewed caffeinated (400 mg; 4.1 ± 0.5 mg·kg^−1^) or placebo gum for 5 min before post-intervention testing, and cognitive performance outcomes were unaffected by caffeinated gum. Caffeinated gum has been shown to facilitate peak plasma caffeine levels (44–80 min) faster than encapsulated caffeine (84–120 min) [71], although Russell et al. [44] focused on the assessment of hormonal markers. Future caffeine and mental fatigue research could assess plasma and salivary caffeine levels to open discussion on the interaction of the physiological mechanisms of caffeine and mental fatigue.

Overall, six effect sizes from five studies showed that the effect of caffeine on reaction time was not significantly different to that of placebo. However, three studies had a positive effect [44,54,57], one showed a negative effect [55], and one showed a null effect [61]. The comparison of results between studies was limited by the use of the inclusion of exercise protocols plus caffeine administration with different forms (e.g., beverages, gum, and capsules), dosages, and timings. Recently, it was posited that perceived mental fatigue is different to physical, tiredness, stress, mood, and motivation [72]. Although the strategies differ greatly, future research could compare the effectiveness of caffeine beverage-, gum-, and capsule-based strategies on cognitive performance in mentally fatigued individuals, after a mentally fatiguing intervention alone, to highlight the nutritional strategy with the greatest potential to mitigate mental fatigue specifically.

### 4.3. Carbohydrate

Two studies were analyzed regarding the effect of carbohydrate on both accuracy and reaction time [59,60], with two studies included in the analysis related to reaction time [63,64]. Lee et al. [64] observed the cognitive performance of 12 participants in five cognitive tasks (symbol digit matching, choice reaction time test, PVT, digit span, and search and memory) before and after a 75 min treadmill run at 70% VO_2max_ with induced hyperthermia. Then, they consumed 1 mL·kg^−1^ of a 6.8% CHO solution or placebo at the start, every 15 min during exercise and between cognitive tests after exercise. Carbohydrate improved working memory in the digit span task compared with placebo, although there were no effects in the other tasks. It has been recommended that players drink highly concentrated (12% carbohydrate) beverages during matches with limited drinking opportunities [73]. Aligned to the recommendations of Rodriguez-Giustiniani et al. [73], one study included 15 soccer players who consumed a 500 mL 12% carbohydrate–electrolyte (60 g carbohydrate + 205 mg sodium), placebo–electrolyte, or water beverage before (250 mL) and at the half-time interval (250 mL) of a 90 min soccer match simulation [59]. Cognitive performance was assessed using the Computerized Mental Performance Assessment System (COMPASS, Northumbria University, UK). Pre- to post-exercise comparisons showed that reaction time was faster in all tests and conditions, whereas the effects on accuracy were mixed. Despite the raw data showing some changes, all effects were insignificant. The results from the studies of Lee et al. [63] and Harper et al. [59] highlight the importance of the selection of cognitive tasks for the assessment of cognitive performance when drawing inferences from data. Harper et al. [59] posited that their results supported previous research that suggested mental fatigue impaired soccer-specific performance [23]. However, Harper et al. [59] did not measure mental fatigue, and there were no statistical differences. It is unclear whether improved reaction time and accuracy in computerized cognitive testing batteries translates to improved soccer performance, and future research designs could be strengthened by considering the practical usefulness of such data before imposing extensive cognitive testing procedures on large squads of athletes. Crossover studies could be designed to the compare the effects of induced mental fatigue appropriately defined cognitive domains [33], after appropriately measured markers of mental fatigue (e.g., visual analog scales), on cognitive performance in computerized tasks and sport-specific decision-making tasks, to deduce the transferability of results.

The effects of carbohydrate mouth-rinsing on post-exercise Stroop task performance was assessed by Konishi et al. [60]. A solution of 6.4% maltodextrin or water (control) was rinsed for 5 s approximately every 10 min during 65 min of exercise at 75% VO_2max_. The results of this meta-analysis showed a positive effect of the intervention, supporting the data showing that reaction time significantly increased (i.e., was slower) from pre- to post-exercise in the control condition (529 ± 45 vs. 547 ± 60 ms, *p* = 0.029) but not the carbohydrate condition (531 ± 54 ms vs. post-exercise 522 ± 80 ms). Carbohydrate mouth-rinsing has been shown to activate the oral receptors and subsequently stimulate activity at prefrontal cortex and anterior cingulate cortex in rested conditions, and the same could be said when performing exercise. Given that the prefrontal cortex and anterior cingulate cortex are implicated in mental fatigue and cognitive domains such as executive function, working memory, and processing speed, measuring the activity of these brain regions could offer novel insights to strengthen the quality of mental fatigue and carbohydrate research. Future research could assess brain activity before and after a mentally fatiguing intervention, as well as following subsequent exercise, to identify whether carbohydrate mouth-rinsing effectively stimulates activity at brain regions associated with mental fatigue.

Overall, this meta-analysis showed that reaction time was unaffected by carbohydrate-based strategies. Using specifically designed cognitive testing batteries could capture the holistic effects of interventions on different cognitive domains in future studies; however, the potential for saturation through assessment of performance in too many cognitive performance tasks should be considered. In the future, researchers could carefully select the cognitive tasks used to assess cognitive performance on the basis of validity, reliability, and alignment to a defined cognitive domain with a neuropsychological underpinning, such as those of Harvey [33] or Cattrell–Horn–Carroll [32], provided the rationale is clear.

### 4.4. Energy Drinks

Two studies produced four effect sizes [42,65] (three from different interventions in the study of Pomportes et al. [42]), showing that energy drink consumption had the greatest positive effect compared with placebo (i.e., favored the intervention). Having recruited 15 trained cyclists, Lassiter et al. [65] studied the effects of consuming an energy drink (160 mg caffeine, 54 g carbohydrate, taurine, and *Panax ginseng*), compared to those of a placebo, on performance during a reaction time task, tapping task, executive function task (reaction time plus tapping), and Stroop task alongside cycling performance. Participants produced more taps per second in the energy drink condition compared with the placebo condition, although no other effects were found, and the authors noted that exercise appeared to improve performance in the Stroop, reaction time, and executive function task. Similarly, Pomportes et al. (2019) [42] investigated the effect of ingesting an energy drink made up of carbohydrate, caffeine, and guarana on 40 min running performance and cognitive performance (Simon task) in 10 modern pentathlon athletes. All conditions enhanced information processing speed post exercise, and there was an interaction between drink and exercise on reaction time. Energy drinks with similar formulations to those aforementioned may improve cognitive performance and, thus, deserve further exploration. However, aerobic exercise has been shown to improve executive function [70] which likely explains, to some degree, the improvements in cognitive performance. Interpreting the effect of energy drinks is hampered by the inclusion of only two studies, and it is important to note that it is difficult to quantify the relative effects of the several ingredients that were in said energy drinks that potentially enhanced performance. Future research could compare the effects of nutritional interventions and aerobic exercise, in isolation, on cognitive and exercise performance under mentally fatigued conditions to elucidate the relative influence of each respective intervention and in combination to identify whether exercise could have additive ergogenic effects.

## 5. Limitations and Future Directions

The aim of this study was to quantify the effects of nutritional interventions on cognitive performance outcomes that are often measured in mental fatigue research. Limiting the criterion to include mentally fatiguing interventions only would have substantially limited the amount of research available for analysis and discussion. Hence, studies that mentioned mental fatigue and/or included cognitive performance outcomes often measured in mental fatigue research were included. Allowing the inclusion of crossover and parallel-group design studies in the meta-analyses may have skewed results [74]. To mitigate these limitations, a stringent protocol was followed from the screening process to provide information on study quality assessment, which attempted to quantify the effect of an intervention of mental fatigue in sport, military, and aerospace populations. In addition to subjective mental fatigue, numerous top-down based physiological approaches have been used to estimate mental fatigue, including EEG and pupillometry, with primary outcomes comprising performance in computerized tasks requiring attention, mathematical skills, or ecologically valid tasks such as driving simulations [26]. Therefore, the effect sizes from the meta-analysis may have differed had other outcomes been used, such as subjective ratings of mental fatigue measured using a 100 mm VAS. Moreover, although the broader cognitive domain of executive function could have been discussed, it was deemed preferable to discuss accuracy and reaction time the specific aspects of cognition, due to marked differences because of adaptations of established cognitive tasks. In light of this, future research could consider the effect that adapting current cognitive tasks has on the cognitive domain which the original task measures and should consider assigning a cognitive domain using a robust model of cognition such as that of Harvey [33]. Complicating the matter, a plethora of terms have been used interchangeably to describe this construct, such as “self-control strength”, “cognitive load”, and “ego depletion”; however, for the most part, the term “mental fatigue” is commonly used. Attention is turning toward the network physiology of sports and exercise [75] and the interaction of cognitive domains [33]. Future research could align the mechanisms of a given nutritional intervention to those of mental fatigue, in order to identify potentially effective interventions to mitigate the negative physiological and behavioral consequences of mental fatigue. Nutritional interventions excluded from the meta-analysis, due to unavailable or insufficient data, were probiotics [76,77], nitrates [78,79], omega-3 [80], creatine monohydrate [81], and ketone salts [82], which may have led to different results. However, this meta-analysis opens avenues for further research to quantify the effects of interventions on mental fatigue-related outcomes. More high-quality randomized controlled trials need to be conducted by optimizing dietary standardization relating to the type, timing, dosage, and forms of administration of a given nutritional intervention, especially in military and aerospace populations, to create knowledge on the wider practical applications of such interventions.

## 6. Conclusions

This systematic review and meta-analysis attempted to quantify the effects of multiple nutritional interventions on cognitive domains often measured in mental fatigue research. The results suggest that there was no effect of nutritional interventions on accuracy and reaction time in sporting, military, and aerospace populations. Experimental trials that investigate the effects of nutritional interventions on mental fatigue-related outcomes should assess cognitive function using tasks that are assigned to their rightful cognitive domain on the basis of the neuropsychological underpinnings of mental fatigue. Moreover, the standardization of dietary controls, as well as clarification of the fundamental concept of mental fatigue, could foster a greater understanding of the phenomenon in future research. Once the conceptual issues are addressed, next steps might include examining more advanced research methodology, such as, utilizing a complex systems approach, as reductionist efforts to date have not provided a clear understanding of the process from vision to decision. Furthermore, research in this area can be advanced using a transdisciplinary or team science approach as the nature of such a phenomenon requires an understanding of multiple neurobiological systems which spans across scientific disciplines. In addition to physical performance, more high-quality randomized controlled trials are required to elucidate the effects of nutritional interventions commonly used to mitigate the effects of underperformance with cognitive aspects, especially among endurance sports or within extended periods of training, as these areas of research are underrepresented in the scientific literature.

## Figures and Tables

**Figure 1 ijerph-19-00307-f001:**
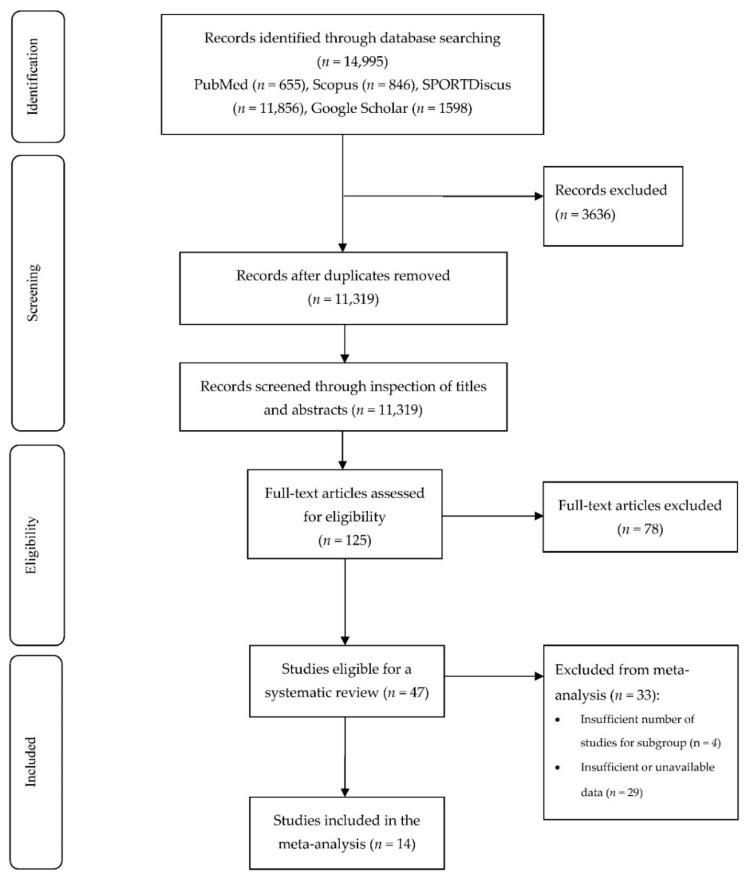
Flowchart of the meta-analysis search process.

**Figure 2 ijerph-19-00307-f002:**
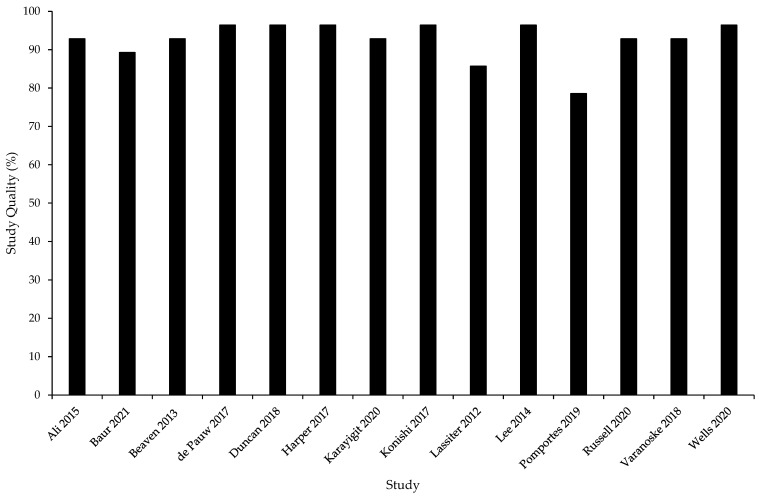
Study quality assessment using an adapted version of the Downs and Black checklist [49].

**Figure 3 ijerph-19-00307-f003:**
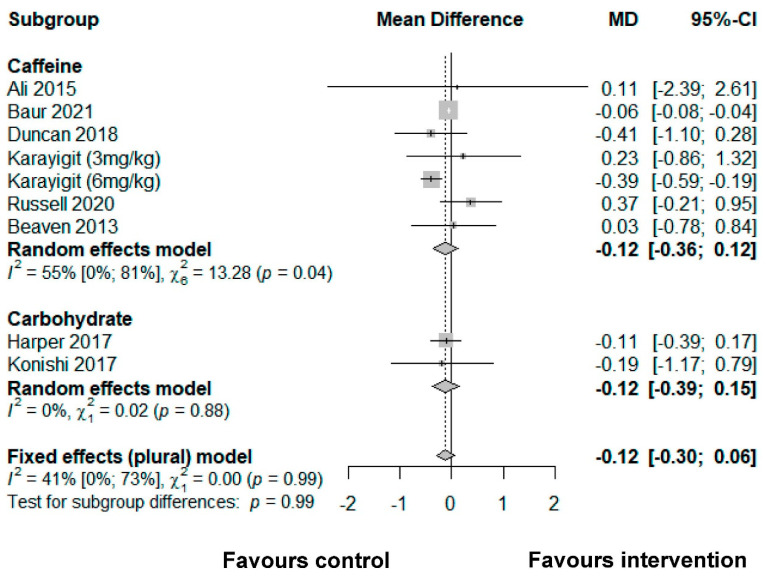
A meta-analysis of the effect of nutrition interventions on accuracy.

**Figure 4 ijerph-19-00307-f004:**
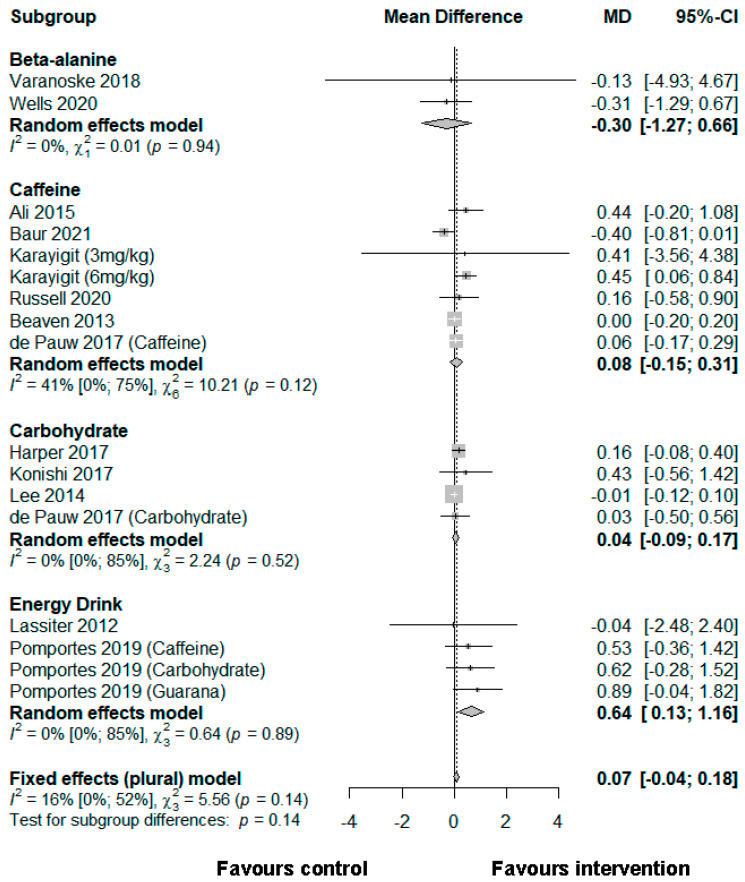
A meta-analysis of the effect of nutrition interventions on reaction time.

**Table 1 ijerph-19-00307-t001:** Search terms and results before deduplication.

Search Terms	PubMed	Scopus	SPORTDiscus	Google Scholar
	Results	Results	Results	Results
(1) “mental fatigue” OR “cognitive fatigue” OR “ego depletion” OR “self-control strength” OR “self-control exertion” OR “mental exertion” OR “cognitive exertion” OR “mental demand” OR “cognitive demand” OR “mental load” OR “cognitive load” AND “sport” AND “exercise” AND “nutrition” AND “intervention”	23	731	614	980
(2) “response inhibition” AND “working memory” OR “executive function” OR “sustained attention” OR “executive attention” OR “emotional regulation” AND “mental fatigue” AND “nutrition”	3	29	1500	294
(3) “dietary intervention” OR supplementation AND “mental fatigue” AND visuomotor OR “physical performance” OR “cognitive performance” AND “sport” AND “ergogenic”	151	58	4386	37
(4) “mental fatigue” AND nutrition AND supplementation AND e-sports athletes OR “military personnel” OR aerospace AND “tactical performance” AND “sustained attention” OR “vigilance test”	378	6	2958	7
(5) “mental fatigue” AND “nutrition” AND “supplement “AND “e-sports” OR “military” OR “aerospace” AND “tactical performance” OR “vigilance” AND “working memory” AND “sustained attention”	100	22	2398	80
Total 14,955	655	846	11,856	1598

## Data Availability

The code that supports results of the meta-analyses is available at: https://osf.io/6rshw/ accessed on 1 October 2021.

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
