# Peer review of "Effects of Nutritional Interventions on Accuracy and Reaction Time with Relevance to Mental Fatigue in Sporting, Military, and Aerospace Populations: A Systematic Review and Meta-Analysis"

_ijerph, 2021, doi:10.3390/ijerph19010307_

Round 1
Reviewer 1 Report
The topic is very interesting.
The meta-analysis has been recorded on PROSPERO? Authors should enter the registration code in the methodology section.
A flowchart of the literature article selection process and a table with keywords used as a search strategy should both be integrated into the methodology section.
It would be better to convert table 2 into a bar graph to show the quality of the selected studies.
The paper is structurally well done and the content is also good. Please improve these noted concerns.
Author Response
The topic is very interesting. The meta-analysis has been recorded on PROSPERO? Authors should enter the registration code in the methodology section. A flowchart of the literature article selection process and a table with keywords used as a search strategy should both be integrated into the methodology section. It would be better to convert table 2 into a bar graph to show the quality of the selected studies. The paper is structurally well done and the content is also good. Please improve these noted concerns.
Response: Thank you for clear interest in the area and desire to improve the meta-analysis with these helpful comments. The protocol was not registered on PROSPERO due to the prolonged decision timeline of PROSPERO reviews, which was a warning on the website, due to pressures related to COVID-19. Therefore, it was decided that the protocol be uploaded to Open Science Framework as an alternative (https://osf.io/6rshw/). We hope this clarifies our reasoning and believe this open-source availability of procedures is a suitable alternative. In addition to the flow chart, a table with keywords used as a search strategy has been added to the methodology section. Table 2 has been converted into a bar graph, and Lassiter 2012 and Lee 2014 have been added as appropriate. Finally, a supplementary file is provided to show removals and one adaptation of the Downs and Black checklist:
Page 2, Lines 224-225: Items 9, 13, 16 and 17 were removed and the wording of item 5 was changed to link more closely to the aims of this study (Supplementary file 1).
Reviewer 2 Report
This is a well written manuscript which will likely be well read following some modifications and adjustment of rationale and argument. The focus on mental fatigue could be a real highlight of the manuscript but as is the included studies do not appear to have examined the effect of supplementation on mental fatigue per se, rather they are supplement based studies and the authors are looking to infer what implications such studies have for mental fatigue as a result. This is a little opaque in aim though as readers will expect to see studies on mental fatigue included, yet there are none and the studies included in the review appear to be more supplement studies. Some clarification is needed throughout the review in respect to that point.
In the introduction, final paragraph where the aims are stated, it would be useful to explain what doing the systematic review and meta analysis would achieve, ie what practical use would it be. It is fine to say that this topic ha snot bene reviewed before but what is the outcome? Better advice for scientists and practitioners working in that sector? Providing guidelines for use of nutritional supplements in different settings? Stating this would add context to the ‘why’ the authors are doing the review.
Table 1: Im not convinced this is really needed if the flow chart is provided.
Figure 1: n=28 were excluded as data were not available. Did the authors attempt to contact the authors of these studies to retrieve the data, if not, they should have. This needs to be documented and this is a relatively high number to excluded if no direct follow up to authors was made.
Table 2. I would suggest including the reference # as well as the year in this table as it provides easier tracking for readers.
On reading Table 2, I then started to look at the individual studies included in the review and am somewhat at odds to how these studies all relate to mental fatigue. The authors need to do some work here to better explain this as a majority of the studies that are included do not examine mental fatigue in any sense. For example the work by Duncan (2018) is a straight caffeine study, there was no aspect of mental fatigue in their design. Similarly, the work by Ali is a pre-post design examining responses before and after exercise with/without supplementation. The same with Karayigit etc. A stronger explanation is needed as to how these constitute ‘mental fatigue’ or their needs to be a reframe of the title and aims.
In regard to energy drinks, do you actually have sufficient studies here to conduct a meta? N = 2 studies would seem too low to make any generalisable conclusions. While it is possible to perform meta analysis on 2 studies, in this case with 4 ES, it is questionable whether this is appropriate. Are the studies similar enough to do this? Have the authors checked the guidance for meta-analysis provided by Cochrane in this respect? http://cccrg.cochrane.org/sites/cccrg.cochrane.org/files/public/uploads/meta-analysis_revised_december_1st_1_2016.pdf
It would also be useful in the discussion around energy drinks to highlight that we do not know all of the ingredients that were in said energy drinks, and thus cannot be clear what substance or mix if substance might lead to the likely enhanced performance suggested by the analysis.
The statements ‘relevant to mental fatigue’ at the start of the discussion and ‘in the context of mental fatigue’ in the conclusion need changing, as the review does not really address anything to do with mental fatigue from my understanding of the studies retrieved, or the authors need to make a much more robust case why the included studies relate to mental fatigue.
References, see Ref #18, needs one of the ‘18’s removing, Ref #1 and #13 appear incomplete
Author Response
This is a well written manuscript which will likely be well read following some modifications and adjustment of rationale and argument. The focus on mental fatigue could be a real highlight of the manuscript but as is the included studies do not appear to have examined the effect of supplementation on mental fatigue per se, rather they are supplement based studies and the authors are looking to infer what implications such studies have for mental fatigue as a result. This is a little opaque in aim though as readers will expect to see studies on mental fatigue included, yet there are none and the studies included in the review appear to be more supplement studies. Some clarification is needed throughout the review in respect to that point.
We have adjusted the focus of paper to highlight that the findings related to cognitive function that have implications on mental fatigue. Our intention was to include studies on mental fatigue and while these studies appeared in our initial search they did not meet the final eligibility criteria.
In the introduction, final paragraph where the aims are stated, it would be useful to explain what doing the systematic review and meta-analysis would achieve, ie what practical use would it be. It is fine to say that this topic has not been reviewed before but what is the outcome? Better advice for scientists and practitioners working in that sector? Providing guidelines for use of nutritional supplements in different settings? Stating this would add context to the ‘why’ the authors are doing the review.
Response: The authors thank the reviewers for suggesting we highlight the broader implications of this impactful study. We have included the following on Page 4, Lines 147-151: Therefore, systematically reviewing and meta-analysing the effects of nutritional interventions on cognitive outcomes typically affected by mental fatigue could support the formation of guidelines surrounding nutritional intake to reduce the risk of suffering cognitive impairment and, thus, inform practitioners’ decision-making in sport, military and aerospace contexts.
Table 1: I’m not convinced this is really needed if the flow chart is provided.
Response: Table 1 has been removed. Thank you for improving clarity in this paper.
Figure 1: n=28 were excluded as data were not available. Did the authors attempt to contact the authors of these studies to retrieve the data, if not, they should have. This needs to be documented and this is a relatively high number to excluded if no direct follow up to authors was made.
Response: Authors were contacted if data was unavailable, and we have clarified the stringency of our procedures. This has been included on Page 2, Line 217. Where data were unavailable after being included in either stage, attempts were made to contact the authors of these studies to retrieve the data.
Table 2. I would suggest including the reference # as well as the year in this table as it provides easier tracking for readers.
Response: Reference numbers have been included in the table. Thank you for helping us improve the readability of the table and article overall.
On reading Table 2, I then started to look at the individual studies included in the review and am somewhat at odds to how these studies all relate to mental fatigue. The authors need to do some work here to better explain this as a majority of the studies that are included do not examine mental fatigue in any sense. For example the work by Duncan (2018) is a straight caffeine study, there was no aspect of mental fatigue in their design. Similarly, the work by Ali is a pre-post design examining responses before and after exercise with/without supplementation. The same with Karayigit etc. A stronger explanation is needed as to how these constitute ‘mental fatigue’ or their needs to be a reframe of the title and aims.
Response: Thank you for pointing out the lack of clarity in our language. We have elaborated on this throughout, such as by changing the wording from “relevant” to “cognitive performance outcomes often measured in mental fatigue research”. In addition, we have included the following on Page 5, Lines 600-606:
The aim of this study was to quantify the effects of nutritional interventions on cognitive performance outcomes that are often measured in mental fatigue research. Limiting the criterion to include mentally fatiguing interventions only would have severely limited the amount of research available for analysis and discussion. Hence, it was decided to include any studies that mentioned mental fatigue and/or included cognitive performance outcomes often measured in mental fatigue research.
In regard to energy drinks, do you actually have sufficient studies here to conduct a meta? N = 2 studies would seem too low to make any generalisable conclusions. While it is possible to perform meta analysis on 2 studies, in this case with 4 ES, it is questionable whether this is appropriate. Are the studies similar enough to do this? Have the authors checked the guidance for meta-analysis provided by Cochrane in this respect? http://cccrg.cochrane.org/sites/cccrg.cochrane.org/files/public/uploads/meta-analysis_revised_december_1st_1_2016.pdf. It would also be useful in the discussion around energy drinks to highlight that we do not know all of the ingredients that were in said energy drinks, and thus cannot be clear what substance or mix of substance might lead to the likely enhanced performance suggested.
Response: Whilst we did meta-analyse the effect of energy drinks from two studies, we wish to emphasise the limitations of including only two studies in a meta-analysis. For example, the Cochrane handbook states: “Two studies is a sufficient number to perform a meta-analysis, provided that those two studies can be meaningfully pooled and provided their results are sufficiently ‘similar’.” The globally used term “energy drinks” suggests similarity however we do wish to emphasise that there are likely large, immeasurable differences between the ingredients and their relative effects. We have attempted to reiterate this as follows, on Page 4, Line 592 – 595.
Interpreting the effect of energy drinks is hampered by the inclusion of only two studies, and it is important to note that it is difficult to quantify the relative effects of the several ingredients that were in said energy drinks that potentially enhanced performance.
The statements ‘relevant to mental fatigue’ at the start of the discussion and ‘in the context of mental fatigue’ in the conclusion need changing, as the review does not really address anything to do with mental fatigue from my understanding of the studies retrieved, or the authors need to make a much more robust case why the included studies relate to mental fatigue.
Response: Thank you for suggesting this subtle change in language that will markedly improve the understanding of the paper. In the discussion, and throughout the document to match, we have changed the wording from “relevant” to “cognitive performance outcomes often measured in mental fatigue research”. For example, in the discussion, on Page 5, Lines 641–642.
See Ref #18, needs one of the ‘18’s removing, Ref #1 and #13 appear incomplete.
References have been corrected. We thank the reviewer for ensuring precision.
Reviewer 3 Report
In this meta-analysis, the authors examined the impact of nutrition on mental fatigue-induced cognitive impairment. The authors concluded that consuming energy drink, but not beta-alanine, caffeine, and carbohydrate, may improve mental fatigue-induced cognitive impairment.
The authors state “mental fatigue is referred to as a psychobiological state resulting from prolonged periods of demanding cognitive ability” (P1, L39-40). Meanwhile, the authors state “None of the studies identified used mentally fatiguing interventions, but all mentioned and/or were deemed relevant to mental fatigue” (P9, L277-279). This is a big issue for this systematic review because the authors included the study about the relationship between nutrition and exercise fatigue (central motor fatigue)-induced cognitive impairment. The physiological meaning between central fatigue during exercise and mental fatigue is a big difference, thereby being the different impact of nutrition on both.
Minor
The introduction seems too long. Please summarize and make a clear rationale.
Author Response
In this meta-analysis, the authors examined the impact of nutrition on mental fatigue-induced cognitive impairment. The authors concluded that consuming energy drink, but not beta-alanine, caffeine, and carbohydrate, may improve mental fatigue-induced cognitive impairment. The authors state “mental fatigue is referred to as a psychobiological state resulting from prolonged periods of demanding cognitive ability” (P1, L39-40). Meanwhile, the authors state “None of the studies identified used mentally fatiguing interventions, but all mentioned and/or were deemed relevant to mental fatigue” (P9, L277-279). This is a big issue for this systematic review because the authors included the study about the relationship between nutrition and exercise fatigue (central motor fatigue)-induced cognitive impairment. The physiological meaning between central fatigue during exercise and mental fatigue is a big difference, thereby being the different impact of nutrition on both.
Response: The authors thank the reviewer for helping us to make the aims and eligibility criteria of the analysis clear to the reader. To address this comment, and the comments of other reviewers, we have changed the wording from “relevant” to “cognitive performance outcomes often measured in mental fatigue research”. For example, in the abstract (Page 1, Line 23) and introduction (Page 3, Lines 107–108).
Minor - The introduction seems too long. Please summarize and make a clear rationale.
Response: Thank you for aiding us in summarizing the rationale for our work. The following pieces of writing have been removed to summarize the information:
Previously Page 2, Lines 73-74: Recently, studies have monitored the physiological expression of mental fatigue.
Previously Page 3, Lines 97-99: Executive functioning controls all other cognitive abilities, such that cognitive resources can be effectively utilized to solve problems efficiently, and plan for the future
Previously Page 3, Lines 110–112: Efficacious interventions that serve to protect cognitive domains such as accuracy and reaction time would be of value to practitioners in sport and exercise, military, and aerospace contexts.
Previously Page 3, Lines 133–137: Research has also found that the consumption of caffeinated [46] and caffeine-free [47] polyphenol beverages attenuated increases in perceived fatigue and decreases in alertness during cognitively demanding computerized tasks, suggesting that the active ingredients may be beneficial in isolation or synergistically.
Previously Page 3, Lines 140-142: notwithstanding the different objective and subjective performance outcomes used to assess the impact of a given nutritional intervention.
Round 2
Reviewer 2 Report
The authors have done a good job responding to the comments raised. The changes made add to clarity of the review in my opinion. I have no further suggestions to make in relation to the paper and believe it is in acceptable format.
Reviewer 3 Report
Well done, I have no further comments.